# Younger Women with Lipedema, Their Experiences with Healthcare Providers, and the Importance of Social Support and Belonging: A Qualitative Study

**DOI:** 10.3390/ijerph20031925

**Published:** 2023-01-20

**Authors:** Vilde Christoffersen, Merete Kolberg Tennfjord

**Affiliations:** Department of Health and Training, Kristiania University College, 0152 Oslo, Norway

**Keywords:** lipedema, Norway, qualitative study, women, belonging, social support, healthcare providers, relationships

## Abstract

Lipedema is a chronic adipose tissue disorder affecting approximately 11% of women worldwide. The illness is often misdiagnosed as obesity, and because of this, women often struggle in meetings with healthcare providers. Few studies have assessed these encounters of younger women with lipedema. The aims of this qualitative study were to explore women’s experiences in meetings with healthcare providers and the importance of social support and belonging, with a focus on younger women. Fifteen women with lipedema between the ages of 21 and 47 years (mean age 36.2 years) were interviewed. The results indicated that women felt stigmatized by healthcare providers and that younger women in their 20s and early 30s struggled more often than women of higher age when receiving their diagnosis. The feeling of shame and stigma were also dependent on the woman’s resources in handling the illness. The younger women reported that their self-confidence and romantic relationships were challenging. Social support and the feeling of belonging through romantic relationships or support groups were important resources for managing the illness. Highlighting the experiences of women may aid in increasing recognition and knowledge of lipedema. This in turn may reduce the stigma and lead to equitable healthcare services.

## 1. Introduction

Lipedema is a chronic adipose tissue disorder [1] that almost exclusively affects women. Its cause is unknown, but it is thought to be hereditary; the disorder may also be hormone-dependent, and it includes changes in adipose cell function and vascularity abnormalities [2,3]. Furthermore, the exact prevalence is uncertain, but it is estimated that up to 11% of women worldwide suffer from this disorder [4]. However, the number is probably underestimated due to frequent underdiagnosis. Few diagnostic criteria of lipedema exist, but common characteristics are unusual symmetrical fat deposits in the buttock, thigh, abdomen, or arms [1], which often cause pain and tenderness and may result in easy bruising and constriction of movement. Furthermore, a typical woman with lipedema has a higher body mass index (BMI) than average, and this enlargement is persistent despite weight loss. There is also a disproportion of adipose tissue between the upper and lower body halves, with minimal involvement of the feet. Minimal pitting edema is also almost exclusively found in this group of women [3]. Four stages of lipedema that represent disease severity have been presented, ranging from the least severe stage (stage I) up to lipolymphedema (stage IV) [1].

Due to the association of lipedema with a higher BMI [3], women with lipedema are often misdiagnosed as obese. Further, the progression of the disease is linked with the progression of obesity [3]. As a result, the affected women can feel rejected by healthcare providers due to stigmatization [5]. This stigma and the changes in the body may further impact quality of life [5] and lead to self-stigmatization, stress, anxiety, and feelings of shame and guilt [6]. Stigma can be described as an attribute that conveys devalued stereotypes and may be seen as profoundly discrediting [7].

There is currently no cure for lipedema, and the main goal of the treatment is symptom relief and self-care management [3]. Most importantly, the management of lipedema should be multifactorial and of a holistic character, including psychological and emotional aspects [3,8]. Conservative management of lipedema includes diet and physical activity. Although the efficacy of these interventions is unclear when it comes to the removal of the adipose tissue, there is no doubt that weight control or even weight loss is beneficial for the general health and well-being of these women [3]. Other management alternatives for symptom relief involve decongestive lymphatic therapy and compression [3]. When conservative management fails, surgery including liposuction and, in complicated and advanced stages of lipedema, debulking surgery may be offered [3].

Misconceptions about the disorder, the uncertainty of its diagnosis, the lack of effective conservative treatment options, and the lack of reliable and valid outcome measures to assess the efficacy of treatment may cause affected women to experience physical and psychological distress [2,3,4,8]. As a consequence, women with lipedema often end up searching for answers themselves [9]. In a recent qualitative study, women aged 30–60 years were asked about their experience of living with lipedema [10]. Many of these reported how pain affected their daily lives and their thoughts about themselves. Furthermore, the women felt criticized about their weight and did not receive supportive advice and professional care.

Few studies have focused on younger women (women in their 20s and early 30s) with lipedema [5,9,10,11,12], and as far as we know, no qualitative studies have explored younger women meeting healthcare providers and the roles of social support and belonging among them. Younger women may experience a different burden of disease as compared to women of higher age due to a shorter period over which pain has impacted their work and social lives [12]. Younger women may also be in a period of life where social acceptance and approval are important societal factors due to a constant search for validation across both social media and in life in general. This may further impact their search in finding a partner and thus enhance physical and psychological distress. Social support is essential in fulfilling the need to belong, e.g., in friendships, in family or romantic relations, or through work environments. According to Antonovsky [13], social support is one of the necessary resources for handling stressors in life and is therefore an essential aspect of the well-being of women. Social support access and feeling of belongingness in women may therefore be associated with their experience of living with lipedema. Furthermore, it is essential to understand how women experience meetings with healthcare providers so that the issues being faced can be elucidated. This will in turn provide healthcare providers with more insight into how women experience these meetings. Hence, the aims of the present study were to explore how women with lipedema experience meetings with healthcare providers and how the need for social support and belonging may impact their lives living with the disease, with a special focus on younger women.

## 2. Materials and Methods

A qualitative research design was chosen for this study. Qualitative methods are a preferred approach in health research when the aim is to understand more about a phenomenon. These methods involve exploring and providing insights into the experiences, perceptions, behavior, and processes of people and the meanings they attach to them [14]. A phenomenological approach enabled us to examine individuals based on their behavior and subjective perception of reality [15]. The design, analysis, and reporting of the present study followed the Consolidated Criteria for Reporting Qualitative Research [16].

### 2.1. Recruitment and Participants

An announcement concerning recruitment for participation was made public on the Facebook account of the Norwegian Patient Association for Lymphedema and Lipedema. Eligible women were encouraged to contact the primary researcher of the study if they met the following inclusion criteria: had a lipedema diagnosis given by a healthcare professional and were aged 18–50 years. Sixty women responded to the announcement by either email or phone. The primary researcher made contact by email and explained the study details including the exclusion criteria, which were having severe pathology (malignancy), cardiovascular conditions, or immune system diseases diagnosed by a specialist; pregnancy, childbirth within the last 12 months, or breastfeeding; severe psychiatric disorders that demand or had demanded admission to the hospital; or personality disorder diagnosed by a specialist. Applying these exclusion criteria resulted in the identification of 35 women: 20 did not respond by email, 8 were >50 years old, and 7 had additional disorders. The participation criteria were therefore met by 25 of the original 60 women.

There are four healthcare regions in Norway: Northern, Central, Western, and South-Eastern, each of which includes several public hospitals. This division into different healthcare regions may have resulted in differences in the available lipedema treatments. The study therefore included 15 women who represented a geographical spread across all four healthcare regions of Norway, a number thought to be adequate to answer the aims of the present study and in line with the previous study by Melander et al. [10]. Further, we included women that represented the desired age range, with a focus on including women in their 20s and early 30s. Because the number of interested participants exceeded the planned sample size needed for the project, purposeful sampling of the desired participant characteristics was performed, and a waiting list was compiled of those who were not chosen.

### 2.2. Data Collection

An informed-consent statement was received by email prior to the collection of data through telephone interviews. This made it possible to access a greater geographical spread of participants and was also considered favorable since the women might feel more comfortable sharing their experiences while in their homes. The primary researcher (V.C.) performed the interviews in November 2021, and the interviews were recorded using a secure dictaphone application (Nettskjema: a Norwegian internet-based tool for data collection).

A semistructured interview guide that asked open-ended questions was used to obtain information about the experiences of the women. The interview guide was developed by discussing the questions with a key person from the Norwegian Patient Association for Lymphedema and Lipedema to ensure that the questions were relevant and understandable. This key person was diagnosed with lipedema and considered suitable for this task. Questions were asked in four main categories: social background, the experience of living with lipedema, social support, and treatment options. The order of the topics was chosen to guide the participants toward knowledge of the topic [17]. If an answer was unclear or did not provide sufficient information, questions such as “how did this affect you?” and “could you please give me an example?” were asked to clarify their experiences. At the end of the interview, the women were asked to mention anything that had not been discussed that they felt was lacking. Each interview took 20–50 min.

### 2.3. Analysis

The data were transcribed verbatim. The researchers applied a qualitative, phenomenological approach to the analysis. A theoretical interpretation was performed to systemize the transcripts to implicitly cover the various themes and subthemes. The researchers strived to follow the criteria of trustworthiness reported by Green and Thorogood [17], which included credibility, transferability, dependability, and confirmability. This was ensured by both authors reading the transcripts before discussing alternative interpretations and themes. Furthermore, the authors beforehand clarified potential preconceptions that they brought into the study. Lastly, we strived for a transparent analytical process, following the steps by Brown and Clarke [18].

A thematic analysis was applied to the data to identify, analyze, and report patterns therein, to minimally organize and describe the dataset in detail [18].

The step-by-step guide suggested by Brown and Clarke [18] was implemented to facilitate the analytic process. The following steps were performed by all of the authors: (1) familiarization with the data by repeatedly reading the transcripts, (2) assigning preliminary codes to recognized topics, (3) collating codes into potential themes and collating relevant data, (4) reviewing and validating the identified themes, (5) repeated reading of the themes to ensure the inclusiveness of coding and categorizing for all of the themes in order to validate and determine the credibility of the data analysis, and (6) arranging and reporting the findings by selecting appropriate statements that illustrated the chosen themes. All statements were translated from Norwegian to English and were edited lightly to achieve greater readability while remaining as close to the original statements as possible. Statements were chosen to reflect different age levels while maintaining the focus on the younger women in the sample.

### 2.4. Ethical Considerations

All participating women were informed about the purpose and aims of the study both in writing and orally. They were also informed about their right to privacy, how the collected data were securely stored, that they could withdraw from the study at any time, and that they could choose not to answer any question that they felt would affect them in a negative way. The researcher strived to keep each conversation as safe and comfortable as possible, and encouraged the women to openly and honestly share their experiences of living with the illness at their own pace. Although some women experienced strong emotions during the interview, all of them answered the planned questions. Any identification information was removed, and the women received fictitious aliases. They were also informed that they could make comments about and request corrections to their statements, but none of them did so. The study was approved by the Norwegian Centre for Research Data (approved 14 October 2021; reference 493754). Recordings, transcripts, and signed consent forms were stored according to the General Data Protection Regulations. The study was conducted in accordance with the Declaration of Helsinki.

## 3. Results

The background characteristics of the 15 women included in this qualitative study are presented in Table 1. The mean age of the participating women was 36.2 years (range = 21–47 years). Thirteen had degree-level education, 10 women were either married or in a relationship, and 7 had children.

Two main themes concerning the experiences of women with lipedema emerged from the analysis: (1) how women with lipedema experience meeting with healthcare providers and (2) the importance of social support and the need to belong.

### 3.1. Theme 1. How Women with Lipedema Experience Meeting with Healthcare Providers

Three subthemes emerged from the analysis: (1) “various reactions to receiving a diagnosis”, where the women described their reactions with relief and anger; (2) “misconceptions about the illness”, where the women described being labeled and stigmatized as overweight or obese; and (3) “engagement and knowledge lead to appreciation”, where the women felt that they were seen and heard by their healthcare providers.

#### 3.1.1. Various Reactions to Receiving a Diagnosis

Receiving their diagnoses had been a long and difficult journey, with the women visiting several doctors, physiotherapists, or other healthcare providers. They often felt helpless, not being taken seriously, overlooked, and left to themselves. One woman (Nora, 28 years) said the following: “I think the biggest burden of living with it is just how little knowledge there is in the healthcare system”. However, receiving the diagnosis could be viewed as a double-edged sword. While most of the women felt relieved and were happy about finally getting an answer, there were some younger women who reacted with grief due to them knowing that the illness is chronic and possibly progressive and able to cause greater problems in the future. Amalie (32 years) described receiving her diagnosis as follows: “Ever since I got diagnosed, my life has been dreadful”. Another woman in her early 20s went to a physiotherapist who specialized in lipedema with the aim of denying her suspicions, but was disappointed when she was instead diagnosed. Petra (24 years) expressed her feelings like this: “(…) It was really sad (…). I had my suspicions and I actually went to a physiotherapist thinking: ‘no, of course I don’t have it’”.

The women who were helped and received a diagnosis still felt like they had to educate their primary care doctor about the illness. Some doctors were responsive and were happy to read about and learn more about lipedema, whereas other doctors seemed to ignore the information about the illness provided by the women, which resulted in many women feeling embarrassed or bothered. Many women also felt labeled when they received their diagnosis, and would rather be overweight because they would not be burdened by a diagnosis. Anger reactions were common when the women received their diagnosis, as it confirmed that there was something wrong with their body, and that it was an actual illness rather than just in their head. Lilja (47 years): “It was like all my energy disappeared because I finally understood that there was nothing wrong with my head, and then I got furious because I have endured so much pain for so many years. I was angry and disappointed, and I don’t know how to describe the feeling, but I didn’t have much nice to say about doctors at that time”.

The women reported that they were frequently prescribed pain medications of various potencies, even if they felt the medication had no effect on their symptoms. The pain they had to endure over many years and the subsequent fatigue made them angry at all of the healthcare providers they had met who had not identified or recognized that there was something wrong with their bodies other than being overweight.

#### 3.1.2. Misconceptions about the Illness

When meeting healthcare providers, the women were often labeled as being lazy and asked if they had tried to lose weight. They were frequently asked about their BMI and were told that all their problems would disappear if they lost weight. The women were given weight-loss medications, felt pressured, and were strongly advised to seek weight-management surgery. Amalie (32 years) said the following: “I feel that the doctors put a label on us as overweight, and then they are done with us”. The same feelings of misconceptions were shared by Margrethe (38 years): “I got diet pills. I didn’t ask for it. It is impossible to be as active and live as healthy as I am and to look like this. The doctor asked about my BMI several times and has written in my papers that I am morbidly obese, even though I told him something was wrong”.

Several participants disliked that the diagnosis was considered an “obesity-related” or “fat-related” disease as this implied that the illness was self-inflicted, which they believed caused more stigma. Some of the women had been referred to overweight clinics and dietitians but could not receive help because they did not meet the criteria for obesity. Several of the women claimed that meeting with healthcare providers affected their thoughts about themselves. The women felt that the healthcare providers did not listen when they explained their mental health issues, which made them feel rejected. One woman said the following: Maria (40 years): “I think it is important to be heard because I think a lot of people with lipedema have some mental challenges or poor self-esteem because of it. It’s extra difficult being rejected because you just want to give up”.

Several of the women described that they felt like they were disrespected and not being treated in a professional manner during meetings with specialists. One woman explained in her follow-up consultation how she was informed that she had not been put on the waiting list for lipedema surgery because the doctor did not believe she would manage to lose the required weight. Such examples made many of the women skeptical of consultations with surgeons and healthcare specialists. Furthermore, some of the women even had to travel abroad to receive help after meeting with healthcare providers who did not pay attention to their problems.

#### 3.1.3. Engagement and Knowledge Lead to Appreciation

While many of the participants did not have their needs met, some women explained that they had. This was most common when meeting a physiotherapist with knowledge about lipedema, which led to them receiving noninvasive treatments such as pulsator treatment, manual lymphatic drainage, and compression garments. Margrethe (38 years) said the following: “I´m really lucky I met my physiotherapist. She has knowledge and helps me with lymphatic drainage and compression garments”. Furthermore, the women who were the most satisfied with their primary care providers were also those who felt listened to or heard, and consequently described themselves as lucky. These women also highlighted their feelings of appreciation toward healthcare providers who were eager to learn about the illness.

### 3.2. Theme 2. The Importance of Social Support and the Need to Belong

Differences were identified in how the women were affected and lived their lives with lipedema, which seemed to stem from differences in their surroundings. Two subthemes emerged: (1) “self-confidence and romantic relationships are more challenging at a younger age” and (2) “social support is a necessity”.

#### 3.2.1. Self-Confidence and Romantic Relationships Are More Challenging at a Younger Age

Finding clothing that properly fit their body was a difficult task for all of the women and magnified the previously manifested doubts that they had about their bodies. Regarding self-confidence and body confidence, women in their 20s and early 30s had more negative feelings toward their bodies, describing a feeling of embarrassment. They found it more challenging to explain to friends of similar ages what the illness did to their bodies and that they could not lose weight. In social settings, they had no problems keeping up, except when food was involved since they were on special diets that avoided weight gain. The constant thoughts about food and dieting affected several of the women.

Petra (24 years) said the following: “I do not have the same pain as many others have, but I feel that health care providers have not paid attention to the mental aspect and feel that I am halfway to developing an eating disorder now”.

In addition to the younger women explaining how they felt about their bodies, some women in their late 30s shared that they had felt like this in their younger years. Margrethe (38 years) said the following: “The mental aspect of the illness affected me when I was younger because I had a very tough relationship with food from the age of fifteen to thirty”.

Women in stable and long relationships and those who were married reported that they received support and safety from their partners. These findings were irrespective of their age. They felt that they were loved and appreciated regardless of their illness, and felt that lipedema did not affect their intimacy with their partner. However, even women in stable and long relationships described times of feeling insecure with their partner, and for some women, this was related to times after experiencing pregnancy. This insecurity and embarrassment passed with time. Line (37 years) said the following: “Of-course, you become safer with your partner (…). But yes, that uncertainty and embarrassment has certainly been present”. Some of the younger women and those who were single described that it had been challenging to find a partner because of how they felt about their bodies. Bringing someone home and having to remove compression garments before putting on night garments was described as quite difficult. Cellulite and the shape of their thighs could cause embarrassment, and the pain experienced could lead to a fear of being touched. Their overall confidence levels could be high, but this could plummet when presented with a situation where clothing was expected to be removed. Amalie (32 years) said the following: “Well, to be honest, my self-confidence is ruined”.

#### 3.2.2. Social Support Is a Necessity

The negative feelings the women had towards their bodies and their often low self-confidence highlighted the need for support. Family and friends helped, but having a support person in the same condition as themselves was even better. To receive the support they needed, some of the women had started support groups in their area to help other women with lipedema. Groups like these as well as patient organizations were valued. Some women also found support on Facebook and other social media platforms where women could share their experiences living with the illness and their experiences of encounters with healthcare providers. Isabelle (21 years) said the following: “It’s strange, the amount of support from groups on Facebook. If it hadn’t been for them [support groups] I would never have found someone with similar experiences as me”. Although social support was appreciated by most, several women stated that they eventually had to unfollow these groups because they became triggered by the focus on diet and appearance.

Choosing what to prioritize in life while living with lipedema was challenging for several of the women. Some had to postpone social events, starting a family, and other dreams. As a result, they feared not being accepted due to those choices. Cecilia (43 years) shared her experiences like this: “I don’t have a lot of excess energy and that makes me prioritize, and I kind of chose not to participate in social activities and vacations and such because I don’t endure much and then I kind of chose to work 100%, and for sure it can be discussed if that is the wisest choice, but that’s what I’ve done”.

## 4. Discussion

This study has clearly revealed that there is a lack of knowledge about lipedema, especially among primary healthcare providers, and that this lack of knowledge often leads to misconceptions, disrespect, and the feeling of not being taken seriously by patients. The findings suggest that women in their 20s and early 30s living with lipedema are especially vulnerable, since receiving a diagnosis more often leads to grief compared to the women of higher age in our sample. Another important finding from our study was the importance of support and belonging received from society, from family, or through romantic relationships, which was found for all ages. However, the younger women expressed how their looks and thoughts about themselves caused challenges in their romantic relationships and during intimacy. With respect to theme 1 (“how women with lipedema experience meeting with healthcare providers”), we chose to highlight the salutogenic perspective to showcase the importance of meeting the women with an open mind and a holistic view of the illness. With respect to theme 2 (“the importance of social support and the need to belong”), our findings were discussed from the perspective focusing on the importance of social support and “the need to belong”.

### 4.1. How Women with Lipedema Experience Meeting with Healthcare Providers

Our findings indicate that when the women were seeking a diagnosis, they visited several healthcare providers and were often told that they were overweight or obese [5]. The women found that healthcare providers were not receptive to them presenting their symptoms or diagnosing themselves with lipedema online. Melander et al. [10] reported that women described similar meetings with healthcare providers. Those authors found that women experienced skepticism from healthcare providers when they presented their lipedema symptoms, and often believed that the women tried to blame their looks on someone or something else rather than taking ownership of their condition. This exemplifies stigma among healthcare providers, similar to women in our study often being asked about their BMI and weight in our interviews. The recurring questions about weight and questioning their ability to lose weight could represent weight stigmatization of women that further reduces their self-confidence, which helps to increase their self-stigmatization.

A study by Lawrence et al. [19] on weight bias among healthcare providers supported our findings since they found that many healthcare professionals view obesity as a consequence of poor lifestyle behaviors rather than a disease, which could contribute to the stigma around weight. This weight stigmatization by healthcare providers may have several negative effects, such as leading to inadequate patient assessments, inappropriate diagnoses and treatment decisions, less time being spent with patients, and discharging patients without follow-up [20]. However, Lawrence et al. [19] found that adjusting the way healthcare providers approach and talk to overweight patients can reduce the stigma about weight and improve the patient–provider relationship. Feelings of rejection and being disregarded can increase the degree of self-stigmatization of a patient, which can create or increase the feeling of shame [21]. According to Hoffmann and Tarzinian [22], society expects women to look good and be physically attractive, with good looks equaling good health. Thus, women with lipedema who appear healthy and present with good looks are often neglected or not prioritized when they visit healthcare providers.

Because lipedema can be an invisible illness, the younger women in our study and women in the early stages of lipedema might not have been taken seriously by healthcare providers, therefore experiencing difficulties in being diagnosed. On the other hand, the women of higher age in our study and women in later stages of lipedema may have been met with more stigma due to the illness being more visible. The latter could be related to us finding the women described as being labeled as obese or lazy further reporting that they did not receive appropriate treatment. Since we did not address lipedema stages, we could only speculate if the disease stage could affect the encounters between the women and healthcare providers. This would thus be an important area of future research. This stigma may cause unnecessary harm to the sense of self and worth of women with lipedema [11]. Nevertheless, questions about their BMI status from doctors and other healthcare providers were not meant to be harmful. However, from our results and those of Melander et al. [10] and Dudek et al. [5], it seemed that healthcare providers tend to focus on pathology when meeting with patients. We could argue that this approach might miss important aspects of the reports from the women, since a holistic perspective was not often part of the women’s healthcare encounters in both the present and the latter studies [5,10]. We therefore argue that women who are met with a more salutogenic approach might obtain treatment that is better suited to them. Our findings support this since some of the women were met in a way that made them feel seen and heard. In consultations with healthcare providers, they were given the opportunity to find resources to cope and cooperated well with the providers who were willing to learn about the illness. Consequently, a tendency to only employ a pathological approach when meeting with the women meant that they were met under false pretenses by being labeled as overweight and lazy, and they therefore might not have received the help that they needed.

Puhl and Brownell [23] substantiated this unmet need to receive adequate healthcare by showing that healthcare providers were less happy to treat patients with obesity because they perceived them as being noncompliant, dishonest, and less hygienic. This can contribute to healthcare providers not acknowledging the women and their often poor relationships with food [11]. Pressure from the surroundings, friends, family, and healthcare providers could increase the risk of an eating disorder, and the women in our study felt that this pressure had affected them in social situations. The women were not often believed when they mentioned eating disorders because they showed no visible signs of having such a condition. These misconceptions may stem from people believing that individuals need to look sick or malnourished to be suffering from an eating disorder, and these misconceptions may also apply to healthcare providers [23]. The combined effect of the external pressure to look good and be physically attractive [22] in addition to common misconceptions about the illness could lead to more stigma. Healthcare providers should therefore aim to deconstruct and challenge their ideas about health, obesity, and eating disorders to offer the best possible care to women with lipedema. We believe that integrating a more holistic view is relevant not only to women with lipedema, but also to other diseases that only affect women, especially when the illness has few physical signs of pathology [3].

We found that the women who were prescribed pain medication received little to no effect. The study of Melander et al. [10] produced findings similar to ours, with women being asked to describe their pain and subsequently being prescribed medication based on the interpretation of the doctor. Women have more encounters with healthcare systems during their lifetimes than men do [21]. Lipedema can be painful, and it may be harder for women to receive help because some healthcare providers believe that the pain women experience is largely related to emotions, that it is psychogenic and therefore “not real” [22]. We believe that the patient–provider relationship is an important aspect to consider. Women have personal experiences with and knowledge about their bodies and want to share these with their healthcare provider in order to receive a diagnosis, but they are afraid that they will be misunderstood. The sensitive nature of healthcare encounters can create a power imbalance because of weight stigmatization, which affects the patient–provider relationship [19]. The power held by a healthcare provider may affect whether or not a woman receives a diagnosis, and women with lipedema might therefore not question the decision of their healthcare provider. Furthermore, if the women perceive that they have little influence in these encounters, they may act less competent so that they receive more help and guidance [24]. These factors could contribute to increasing the threshold for seeking help.

Although the women in our study expressed challenges in meetings with healthcare providers, they may have had sufficient coping resources to manage living with their illness. According to Antonovsky [25], resources can include (but not be limited to) some of the following factors: material resources, knowledge and intelligence, ego identity, social support, and preventive health orientation. These coping resources may have only been available to the women in our study because they could afford to travel abroad to seek help. Furthermore, all except one woman were either working or studying, and most were in relationships that could provide them with social support, which was expressed as important when living with lipedema [11]. The study by Melander et al. [10] showed that only 5 out of 15 women were working, their mean age was 46.4 years, and several of the women had lived with symptoms for years prior to diagnosis. Higher age and thus more years of living with lipedema found in the latter study and in the study by Falck et al. (2022) may account for the finding that our study group seemed well functioning in terms of ability to work. A higher educational level has also been associated with a higher level of physical, social, and emotional well-being and functioning [12]. However, although the women seemed to have access to adequate coping resources, several of them expressed that choosing what to prioritize in living with lipedema was challenging, which forced them to make tough decisions about work, social events, and family life. This highlights the importance of women receiving adequate healthcare at an early stage and that young women who present with lipedema symptoms should be taken seriously to ensure that they receive adequate healthcare.

### 4.2. The Importance of Social Support and the Need to Belong

The women who were in stable romantic relationships reported feeling safe and that the illness did not interfere to a great extent with the intimacy with their partner. In contrast, some of the younger women in their 20s and early 30s and those who were not in relationships expressed that it was challenging to be intimate when finding a partner and expressed a lack of self-confidence and self-esteem in situations where they had to undress in front of a potential partner, as they feared how their body would be perceived. Since self-esteem is affected by the three attributes of competence, attractiveness, and likeability, the self-esteem of an individual is related to perceptions of how one is valued by other people [26]. Women who perceive themselves as lacking these attributes could experience lower self-esteem, which could further strengthen the feeling of not belonging or fitting in. This lack of self-esteem could make it even more challenging to approach a potential new partner. The women included in the study by Melander et al. [10] also found it essential to have a close relationship with their partner, yet they felt that their partners did not find them attractive. The women in the latter study found it important to have social support and groups with similar experiences as themselves, which was similar to our findings. This could make the women feel less alone and provide them with information on where to search for help [9]. However, Melander et al. [10] found that social media platforms provide misleading information and promote a one-sided view to women with lipedema. This is comparable to our findings where some of the women expressed that social-media groups can damage the mental health of women by providing misinformation. Despite the latter finding, the women in our study found that these support groups met their need for belonging because they found other women with similar experiences to whom they were able to relate. Close social relationships are preferred since they provide a safe space, but there is evidence that simply being a part of a supportive social network can reduce stress even when the other people do not provide explicit emotional or practical assistance [27]. A sense of belonging among women can increase their happiness and improve their ability to cope with their illness [27]. However, the stigmatization of the illness may make it harder to feel included and valued as a member of a group or society. Because the illness often affects women during particularly vulnerable times of their lives, such as after giving birth and among women in their 20s and early 30s (as found in the present study), it could affect their self-confidence, make it difficult to create social relationships, and hinder their ability to find romantic partners.

Lipedema is accompanied by adverse appearance-related factors that could affect lifestyle changes, such as eating techniques that could lead to eating disorders and social isolation [11]. Our findings suggest that social support has a great impact on how women with lipedema live their lives. Most of the women in our study either worked or studied, and all of them were diagnosed with lipedema and were part of a patient organization. These factors suggest that the women in our study received some degree of social support, which could have met their need to belong.

The strengths of the study are the qualitative design investigating an area of little pre-existing knowledge complemented by the rich narratives from the women included. Another strength is the inclusion of younger women in their 20s and early 30s and their views on encounters with healthcare providers and social support and belonging that have not been explored in previous research. Furthermore, we consider the sample size adequate as no new major themes came up during the last interviews. Thus, saturation was believed to be ensured on the topic of interest. To ensure credibility, both authors read the transcripts and discussed alternative interpretations and themes, clarified potential preconceptions the authors might have had before entering the study, and ensured a transparent analytical process following the steps by Brown and Clarke [18].

The study has some limitations that are important to address. First, we have no data on the time of diagnosis or the stage or severity of the disease, and we may just assume the burden of the disease based on their age. Second, the interviews were performed via telephone, and thus body language could not be observed. However, the advantage of this approach versus face-to-face interviews could be that the women may feel more comfortable being in their private homes and thus be more willing to share their thoughts and feelings. This approach also allowed us to sample from a large geographical spread, thus including the voices of women from the whole of Norway. Lastly, due to most women having a higher level of education and being in either full-time or part-time employment, we may not generalize our results to other women with lipedema or other chronic conditions that may not have the same level of functioning.

This study highlighted the importance of belonging and social support when living with lipedema, and that young women in their 20s and early 30s are especially vulnerable. By showcasing how women with lipedema experience meetings with healthcare providers, we hope that the present study will increase the awareness of the stigma that is present and emphasize the importance of meeting women with respect and understanding. Furthermore, raising awareness about the illness may make healthcare providers diagnose it more frequently. There is clearly a need for specialized healthcare services to take care of these women in a timely and professional manner [3]. However, healthcare providers in primary healthcare settings must be able to recognize common symptoms and refer them for diagnosis. With this focus, women with lipedema may receive help at a younger age and an earlier stage and thereby hopefully prevent years of unnecessary struggle and suffering during meetings with healthcare providers and in general life. Receiving adequate healthcare at an early stage of a disease is also of utmost importance in reducing the burden on society due to expenses associated with disability aid and medical appointments, not to mention the burden on the women themselves. There is a great need to promote the health of women, and performing research on lipedema and other illnesses that affect women can highlight the unmet needs in society and further aim to reduce inequities in the healthcare system. Future research could explore the perspectives of partners and other people close to women with lipedema in terms of their experiences of living close to these women. Furthermore, studies aimed at exploring women´s views and experiences of conservative management would be important to be able to offer treatment alternatives that are both acceptable and tolerable. Lastly, quantifying the economic burden on women living with lipedema across countries would also provide important insight into societal costs, which could identify research priorities and promote a more equitable distribution of healthcare services for women.

## 5. Conclusions

This qualitative study has highlighted how women experience living with lipedema. It became clear that they commonly live with shame and stigma and that the meetings with healthcare providers were often characterized by the provider having a lack of knowledge about lipedema and holding misconceptions about the illness being self-inflicted. It became clear that younger women in their 20s and early 30s struggle more often with feelings of grief when receiving their diagnosis as compared to women of higher age. The younger women also expressed that a lack of self-confidence and self-esteem made them uncomfortable in intimate settings with new partners. On the other hand, women in long and stable relationships felt safe and supported more often, and those with good social support from friends, family, or being a member of a patient organization expressed this as being necessary to cope with their situation.

We hope that by highlighting the experiences of women in meetings with healthcare providers we have contributed to increasing the recognition and acceptance of lipedema, which could reduce the stigma and lead to equitable healthcare services. Furthermore, we believe that the early recognition of young women with lipedema by bringing attention to them will help these women to function better in society, thus reducing the burdens on the women themselves, their close ones, and society as a whole.

## Figures and Tables

**Table 1 ijerph-20-01925-t001:** Background characteristics of the study population, *n* = 15.

Variables	*n* (Range)
Age (years)	36.2 (range 21–47)
Educational level	
Vocational training ^1^	2
Higher level education ^2^	13
Relationship status	
Married or in a relationship	10
Single	5
Women with children	7
Employment	
Full-time or part-time work	10
Student	4
Disability allowance	1

^1^ One woman worked as a chef and one woman worked in sales; ^2^ university or college education.

## Data Availability

The data presented in this study are available on request from the corresponding author.

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
