# Peer review of "Younger Women with Lipedema, Their Experiences with Healthcare Providers, and the Importance of Social Support and Belonging: A Qualitative Study"

_ijerph, 2023, doi:10.3390/ijerph20031925_

Round 1

Reviewer 1 Report

I believe this is a very valuable article that will provide an insight into various aspects of functioning and problems that women with lipedema encounter. 

Despite the fact that the English language is correct in general, I believe the article may be improved with scientific proofreading so that the style is more adequate (academic writing).

Author Response

We sincerely appreciate the Your constructive feedback and have below addressed all comments and suggestions and highlighted the changes using track-changes in the revised manuscript.

Reviewer 1:

I believe this is a very valuable article that will provide an insight into various aspects of functioning and problems that women with lipedema encounter. 

Despite the fact that the English language is correct in general, I believe the article may be improved with scientific proofreading so that the style is more adequate (academic writing).

Author comments: Thank-you for your comment. The paper has already been edited by a professional scientific editing bureau. However, within the short time frame for this review process there was no time for a second round of editing from this agency. If required, we will have the paper edited once more to fit the journal style. 

Reviewer 2 Report

Dear Authors,

thank you for giving me the opportunity to revise the manuscript entitled " Women with lipedema, their experiences with healthcare providers, and their need to belong: a qualitative study". The paper aim  to determine timing and experiece of lipedema in women. The topic is quite interesting and overall can improve the research in field.  Moreover, the topic is fully suitable for the Journal. The manuscript is succinct and well written. Neverthless, there are some critical issue to address:

Introduction: The introduction is lack of information about  lipedema diagnosis. The clinical criteria are the gold standard in this field, but different instrumental tool regarding body composition can help the clinicians. I suggest to read “Dietzel R, Reisshauer A, Jahr S, Calafiore D,  Armbrecht G.  Body composition in lipoedema of the legs using  Dual-Energy-X-Ray-Absorptiometry - a case control study. Br J Dermatol. 2015 Aug;173(2):594-6. doi: 10.1111/bjd.13697”. Moreover, the authors aim to evaluate body composition in POHP, but the rationale of the study is unclear. Please, increase literature about. Moreover, I suggest to add also the possible and effectiveness of different terapuetic approach.

Material and Methods: Line 72-74-please, change the style; 97-103: move the line in the results. Please specify the sample size calculator.

Results: I suggest to add some table resuming the results

Discussion: well done

At the first glance, the paper is extremely lack of literature about. I would have liked to invite the authors to increase this part at the light of the good presented paper. I would suggest: 

"CzerwiÅ„ska M, Teodorczyk J, Hansdorfer-Korzon R. A Scoping Review of Available Tools in Measurement of the Effectiveness of Conservative Treatment in Lipoedema. Int J Environ Res Public Health. 2022 Jun 10;19(12):7124. doi: 10.3390/ijerph19127124. PMID: 35742373; PMCID: PMC9222339." and    " Forner-Cordero I, Forner-Cordero A, Szolnoky G. Update in the management of lipedema. Int Angiol. 2021 Aug;40(4):345-357. doi: 10.23736/S0392-9590.21.04604-6. Epub 2021 Apr 19. PMID: 33870676."

Best Regards

Author Response

Reviewer 2:

We sincerely appreciate the Your constructive feedback and have below addressed all comments and suggestions and highlighted the changes using track-changes in the revised manuscript.

Dear Authors,

Thank you for giving me the opportunity to revise the manuscript entitled " Women with lipedema, their experiences with healthcare providers, and their need to belong: a qualitative study". The paper aim  to determine timing and experiece of lipedema in women. The topic is quite interesting and overall can improve the research in field.  Moreover, the topic is fully suitable for the Journal. The manuscript is succinct and well written. Neverthless, there are some critical issue to address:

Introduction: The introduction is lack of information about  lipedema diagnosis. The clinical criteria are the gold standard in this field, but different instrumental tool regarding body composition can help the clinicians. I suggest to read “Dietzel R, Reisshauer A, Jahr S, Calafiore D,  Armbrecht G.  Body composition in lipoedema of the legs using  Dual-Energy-X-Ray-Absorptiometry - a case control study. Br J Dermatol. 2015 Aug;173(2):594-6. doi: 10.1111/bjd.13697”. Moreover, the authors aim to evaluate body composition in POHP, but the rationale of the study is unclear. Please, increase literature about. Moreover, I suggest to add also the possible and effectiveness of different terapuetic approach.

Author comment: thank-you for the suggestions for literature. We have read the suggested paper, and included more information about diagnostic criteria in the introductory section. However, our study did not aim to assess differences in body composition among women with lipedema, but our aims was to explore how women with lipedema experience meetings with healthcare providers and their need for social support and belonging in living with the disorder. Thus, we have not highlighted the role of body composition in great detail.

We have also included more information about different therapeutic approaches in the introductory section, discussing briefly the strengths and limitations of the various methods.

Material and Methods: Line 72-74-please, change the style; 97-103: move the line in the results.

Author comment: We have changed accordingly

Please specify the sample size calculator.

Author comment: 25 women met the inclusion criteria for this qualitative study. Out of these, 15 women were chosen that best represented the age spread and living spread across the four different healthcare regions in Norway. This number was believed to be adequate to answer the aims of the study and in line with the numbers from a previous study by Melander et al. Furthermore, we believe saturation was met as during the last interviews as no new themes emerged. Information about sample size was added in the methods section 2.1 and under strengths of the study (discussion).

Results: I suggest to add some table resuming the results.

Author comment: We have changed accordingly and included a table with background characteristics

Discussion: well done.

Author comment: Thank-you for your positive comment

At the first glance, the paper is extremely lack of literature about. I would have liked to invite the authors to increase this part at the light of the good presented paper. I would suggest: 

"CzerwiÅ„ska M, Teodorczyk J, Hansdorfer-Korzon R. A Scoping Review of Available Tools in Measurement of the Effectiveness of Conservative Treatment in Lipoedema. Int J Environ Res Public Health. 2022 Jun 10;19(12):7124. doi: 10.3390/ijerph19127124. PMID: 35742373; PMCID: PMC9222339." and    " Forner-Cordero I, Forner-Cordero A, Szolnoky G. Update in the management of lipedema. Int Angiol. 2021 Aug;40(4):345-357. doi: 10.23736/S0392-9590.21.04604-6. Epub 2021 Apr 19. PMID: 33870676."

Author comment: Thank-you for your advice on these papers. We have read the papers and included additional information related to diagnosis of lipedema as well as more information about management of lipedema in the introductory section. We believe these papers gives a good overview of the current literature and have included them in the present paper.

Author Response

Reviewer 3:

We sincerely appreciate the Your constructive feedback and have below addressed all comments and suggestions and highlighted the changes using track-changes in the revised manuscript.

Abstract
• An incidence rate can be given to highlight the potential number of patients experiencing lipedema. By doing this, it will give insight into the rate of lipedema and its inadequate awareness to the readers who are willing to improve their knowledge in this area.

Author comment: We have included information about numbers suffering with lipedema both in the abstract and the introduction.

  • "Fifteen women with lipedema" would be more of a clear indication.

Author comment: we have changed accordingly

  • Belongings to what? Social interpersonal relationships or else? My suggestion is that should be clearly defined both in the abstract and the title.

Author comment: we have tried to explain in the abstract that belongingness was the feeling of belonging to either a person through a relationship, friends, work or even support groups. Since the wording belong and belongingness is frequently used throughout the paper we have kept the wording “belonging” in the title and explained it shortly in the abstract, but further in the introduction. We hope our additional explanation is satisfactory.

Line 29-30
There is also a bunch of conservative methods which can be used to alleviate symptoms such as heaviness and tightness feeling of patients with lipedema. In addition, there is no strict guideline in which liposuction is stressed as the only option for the treatment of lipedema, though I am also aware that there is a bunch of studies in the literature about the process of liposuction in patients with lipedema. Since the
adipose tissue of lipedema quite differs from the adipose tissue at which liposuction can be helpful, this statement should be supported by literature findings. Readers out of this context should be well informed about the treatment and management of lipedema not only from a surgical perspective but also from a conservative treatment
perspective.
Author comment: thank-you for this comment. We agree that there was too little focus into conservative treatment methods in the introduction, and have included additional information in the introduction using recent papers related to the topic. We hope our added information is satisfactory to give readers an overview of currently available treatment methods.

Line 65-66
Is there any chance to clarify more of the aims of this study? Experiences till the diagnosis or multiple visits to healthcare professionals?

Author comment: The aims of the study are explorative in nature aiming to determine women’s experiences during healthcare encounters and the importance of social support and belonging. We therefore believe that the aims should be kept in this open and curious manner. However, in the results section (3.1) we clarified the sub-themes that came up during the various meetings with healthcare providers. These include reactions to receiving a diagnosis, misconceptions about the disease and the importance of engagement and knowledge from the healthcare provider. We hope our answer is satisfactory.

Line 81
Why authors chose a quite narrow range of ages for inclusion? I think it should be justified. Otherwise, I strongly suggest adding the "Younger" word in the title.

Author comment: Due to few studies assessing younger women (women in their 20´s and early 30´s) with lipedema and their experiences with healthcare encounters and the importance of social support and belonging we choose a narrower age range than the previous study by Melander et al. which included women upto age of 60, the studies by Dudek et al. (2016, 2018, 2021) and the study by Falck et al. (2022) We believe that due to younger years, and thus fewer years of living with the disease may affect their level of functioning as well as their views of social support and belonging. We have changed the title accordingly. We have also changed the terms “younger” and “older” throughout the study as we believe these terms were not representative of the population they represented. We have instead highlighted that younger women was in their 20´s and early 30´s and refrained from using “older” but instead described these women of higher age than the youngest. We hope our explanation and reasoning are satisfactory.

Line 88-89
Did patients perform inclusion/exclusion criteria on their own? Or authors also performed a re-evaluation of whether participants met the inclusion and exclusion criteria via any online video conversation? I suggest being clarified on this.

Author comment: We have clarified this issue under part 2.1. The researcher made contact with the 60 women interested in participation and explained the exclusion criteria through email. Of these, 25 women were eligible and met the inclusion criteria of the study. Thus, the 25 women were assessed as eligible after being in contact with the main researcher. We hope our explanation is satisfactory

Line 94
How was this calculation done? Please specify and provide the details of a priori or post-hoc power test results, if applicable.

Author comment: There was no sample size calculation done for this present study. 25 women met the inclusion criteria for this qualitative study. Out of these, 15 women were chosen that best represented the age spread and living spread across the four different healthcare regions in Norway. This number was believed to be adequate to answer the aims of the study and was thus in accordance with the study by Melander et al 2022. Furthermore, we believe saturation was met as during the last interviews no new themes emerged. We have included more details related to strengths of the study, in the discussion section.

Line 99-101
I strongly suggest to the authors add information on cut-off value about how the authors chose to classify patients whether they were older or younger.

Author comment: Thank-you for this comment. We agree that these terms were not adequately addressed and explained in the manuscript. We have chosen to describe the youngest women in the sample as «women in their 20´s and early 30´s» as these were in the age group that differed to women in their late 30´s and higher. We hope our rationale are satisfactory.

Line 113
Should be specified whether this “key person" has lipedema or not, as well as whether she/he has any academic background.

Author comment: the “key” person had the diagnosis of lipedema and was at the time of the study the leader of the patient organization. Although we have no details of her educational background, we consider her feedback as relevant and suitable due to her experience of living with the condition for years. We have added information about this under method section 2.2.

Line 159-160
Were there any previous experiences (of authors) and/or investigations related to the definition of those themes? If so, please specify.

Authors comment: The authors beforehand clarified any potential preconceptions they may have had that could have influenced the data sampling, the analytical process and the definitions of themes. We have added more details in the methods (section 2.3) and under discussion “strengths and limitations”. Further, the authors had at the time of submission already included details of the analytical process according to Brown and Clarke. This process illustrates six steps starting from reading the transcripts to discovering relevant themes from these transcripts. The information is described under 2.3 analysis.

Line 217-219
Is this surgery directly related to affected extremities or some obesity-related surgery?
Should be specified.
Authors comment: The surgery was related to the affected extremities. We have clarified this issue in the results section.

Line 286
I have understood that all patients had moderate to hard levels of difficulty while seeking the reason for their illnesses. Yet, saying this kind of strictly that there is no knowledge about lipedema among all healthcare providers might be a bit harsh.
Authors may think to focus on people who have the capabilities of diagnosing in the primary care setting.

Author comment: Thank-you for your comment. We have added information in the implication section discussing the need for more specialized healthcare services taking care of women with lipedema. However, we believe that health care personnel in the primary care setting in general should be aware of the disorder in order to refer patients they suspect having lipedema. In turn this might impede the process from symptom debut to diagnosis.

Line 333
The main flaw of this qualitative study might be this one. Since the stage of lipedema is mostly based on BMI and their linear relationships with each other, thereby
healthcare providers might think the main problem was being overweight instead of specific clinical findings.

Author comment: Our study results show that most women felt stigmatized due to the misconceptions about the disease from their primary healthcare providers. We have discussed a need for a more holistic view when meeting patients, due to any cause, including being overweight or having lipedema. Although not all healthcare providers had limited knowledge and were disrespectful in our study, the majority of our sample reported not been taken seriously. We believe these findings were already well addressed in the discussion section and followed up with added information in the implications of the importance of awareness and early recognition to avoid the potential detrimental effects this disorder may have on women. We hope this was a satisfactory answer to your comment.

Line 398
Repetitive and specific citation to the Melander et al. Throughout the text, the authors chose to cite to the Melander et al persistently. My opinion, literature findings should be diversified to enhance the manuscript’s scientific integrity irrespective of the
studies whether they are qualitative of quantitative works. Also, there is a lack of characteristics of patients, such as whether was there any conservative treatment among patients if it was performed or taken. The generalizability of the results might therefore be a bit disputable.
Author comment: We acknowledge all studies investigating lipedema; however, we chose to focus on the study by Melander et al due to its qualitative design. However, we have included the studies by Dudek et al 2016, 2018, 2021 and Falk et al 2022 addressing aspects of quality of life through quantitative studies in the discussion supporting the scientific integrity of the present study. Unfortunately, we did not include data that described women’s practice or experiences with conservative treatments. Although this would be an interesting path to follow, this current study did not aim to assess their specific treatments, rather their general encounters with healthcare providers. However, as presented under result women meeting physiotherapists with knowledge, interest and understanding was appreciated and the women described themselves as lucky. We included this focus as future research recommendations. We have included as a limitations that the study results may not be generalized to other women with lipedema and similar populations with a lower educational level and those not being able to work.

Line 404
Compared to what or whom? Since the maximum age was set as 50, I assume being 50 is not directly accepted as being old. Please specify

Author comment: The terms “old” and “young” in this present study is both confusing and not appropriately address and we do apologize for this unclarity. This specific sentence was referred to the patients in the study by Melander et al and that the women in our study was both at younger age and thus had fewer years of living with lipedema. We have refrained from the terms “old” and “young” and rephrased accordingly. Thus when addressing younger participants we refer to them as women in their 20´s and early 30´s.

Round 2

Reviewer 2 Report

All concerns have been addressed

Best